# ALPHA-DIVERGENCE BRIDGES MAXIMUM LIKELIHOOD AND REINFORCEMENT LEARNING IN NEURAL SEQUENCE GENERATION

## ABSTRACT

Neural sequence generation is commonly approached by using maximum-likelihood (ML) estimation or reinforcement learning (RL). However, it is known that they have their own shortcomings; ML presents training/testing discrepancy, whereas RL suffers from sample inefficiency. We point out that it is difficult to resolve all of the shortcomings simultaneously because of a tradeoff between ML and RL. In order to counteract these problems, we propose an objective function for sequence generation using $\alpha$-divergence, which leads to an ML-RL integrated method that exploits better parts of ML and RL. We demonstrate that the proposed objective function generalizes ML and RL objective functions because it includes both as its special cases (ML corresponds to $\alpha \to 0$ and RL to $\alpha \to 1$). We provide a proposition stating that the difference between the RL objective function and the proposed one monotonically decreases with increasing $\alpha$. Experimental results on machine translation tasks show that minimizing the proposed objective function achieves better sequence generation performance than ML-based methods.

## 1 INTRODUCTION

Neural sequence models have been successfully applied to various types of machine learning tasks, such as neural machine translation (Cho et al., 2014; Sutskever et al., 2014; Bahdanau et al., 2015), caption generation (Xu et al., 2015; Chen & Lawrence Zitnick, 2015), conversation (Vinyals & Le, 2015), and speech recognition (Chorowski et al., 2014; 2015; Bahdanau et al., 2016). Therefore, developing more effective and sophisticated learning algorithms can be beneficial.

Popular objective functions for training neural sequence models include the *maximum-likelihood* (ML) and *reinforcement learning* (RL) objective functions. However, both have limitations, i.e., training/testing discrepancy and sample inefficiency, respectively. Bengio et al. (2015) indicated that optimizing the ML objective is not equal to optimizing the evaluation metric. For example, in machine translation, maximizing likelihood is different from optimizing the BLEU score (Papineni et al., 2002), which is a popular metric for machine translation tasks. In addition, during training, ground-truth tokens are used for the predicting the next token; however, during testing, no ground-truth tokens are available and the tokens predicted by the model are used instead. On the contrary, although the RL-based approach does not suffer from this training/testing discrepancy, it does suffer from sample inefficiency. Samples generated by the model do not necessarily yield high evaluation scores (i.e., rewards), especially in the early stage of training. Consequently, RL-based methods are not self-contained, i.e., they require pre-training via ML-based methods. As discussed in Section 2, since these problems depend on the sampling distributions, it is difficult to resolve them simultaneously.

Our solution to these problems is to integrate these two objective functions. We propose a new objective function $\alpha$-DM ($\alpha$-divergence minimization) for a neural sequence generation, and we demonstrate that it generalizes ML- and RL- based objective functions, i.e., $\alpha$-DM can represent both functions as its special cases ($\alpha \to 0$ and $\alpha \to 1$). We also show that, for $\alpha \in (0, 1)$, the gradient of the $\alpha$-DM objective is a combinations of the ML- and RL-based objective gradients. We apply the same optimization strategy as Norouzi et al. (2016), who useed importance sampling, to optimize this proposed objective function. Consequently, we avoid on-policy RL sampling which

suffers from sample inefficiency, and optimize the objective function closer to the desired RL-based objective than the ML-based objective.

The experimental results for a machine translation task indicate that the proposed $\alpha$-DM objective outperforms the ML baseline and the reward augmented ML method (RAML; Norouzi et al., 2016), upon which we build the proposed method. We compare our results to those reported by Bahdanau et al. (2017), who proposed an on-policy RL-based method. We also confirm that $\alpha$-DM can provide a comparable BLEU score without pre-training.

The contributions of this paper are summarized as follows.

- We propose the $\alpha$-DM objective function using $\alpha$-divergence and demonstrate that it can be considered a generalization of the ML- and RL-based objective functions (Section 4).
- We prove that the $\alpha$-DM objective function becomes closer to the desired RL-based objectives as $\alpha$ increases in the sense that the upper bound of the maximum discrepancy between ML- and RL-based objective functions monotonically decreases as $\alpha$ increases.
- The results of machine translation experiments demonstrate that the proposed $\alpha$-DM objective outperforms the ML-baseline and RAML (Section 7).

## 2   COMPARING OBJECTIVE FUNCTIONS

In this section, we introduce ML-based and RL-based objective functions and the problems in association with learning neural sequence models using them. We also explain why it is difficult to resolve these problems simulataneously.

**Maximum-likelihood**   An ML approach is typically used to train a neural sequence model. Given a context (or input sequence) $x \in \mathcal{X}$ and a target sequence $y = (y_1, \ldots, y_T) \in \mathcal{Y}$, ML minimizes the negative log-likelihood objective function

$$\mathcal{L}(\theta) = -\sum_{x \in \mathcal{X}} \sum_{y \in \mathcal{Y}} q(y|x) \log p_\theta(y|x), \tag{1}$$

where $q(y|x)$ denotes the true sampling distribution. Here, we assume that $x$ is uniformly sampled from $\mathcal{X}$ and omit the distribution of $x$ from Eq. (1) for simplicity. For example, in machine translation, if a corpus contains only a single target sentence $y^*$ for each input sentence $x$, then $q(y|x) = \delta(y - y^*)$ and the objective becomes $\mathcal{L}(\theta) = -\sum_{x \in \mathcal{X}} \log p_\theta(y^*|x)$.

ML does not directly optimize the final performance measure; that is, training/testing discrepancy exists. This arises from at leset these two problems:

(i) **Objective score discrepancy.** The reward function is not used while training the model; however, it is the performance measure in the testing (evaluation) phase. For example, in the case of machine translation, the popular evaluation measures such as BLEU or edit rate (Snover et al., 2006) differ from the negative likelihood function.

(ii) **Sampling distribution discrepancy.** The model is trained with samples from the true sampling distribution $q(y|x)$; however, it is evaluated using samples generated from the learned distribution $p_\theta(y|x)$.

**Reinforcement learning**   In most sequence generation task, the optimization of the final performance measure can be formulated as the minimization of the negative total expected rewards expressed as follows:

$$\mathcal{L}^*(\theta) = -\sum_{x \in \mathcal{X}} \sum_{y \in \mathcal{Y}} p_\theta(y|x) r(y, y^*|x), \tag{2}$$

where $r(y, y^*|x)$ is a reward function associated with the sequence prediction $y$, i.e., the BLEU score or the edit rate in machine translation. RL is an approach to solve the above problems. The objective function of RL is $\mathcal{L}^*$ in Eq. (2), which is a reward-based objective function; thus, there is no objective score discrepancy, thereby resolbing problem (i). Sampling from $p_\theta(y|x)$ and taking the expectation with $p_\theta(y|x)$ in Eq. (2) also resolves problem (ii). Ranzato et al. (2016) and Bahdanau et al. (2017)

directly optimized $\mathcal{L}^*$ using policy gradient methods (Sutton et al., 2000). A sequence prediction task that selects the next token based on an action trajectory $(y_1, \ldots, y_{t-1})$ can be considered to be an RL problem. Here the next token selection corresponds to the next action selection in RL. In addition, the action trajectory and the context $x$ correspond to the current state in RL.

RL can suffer from sample inefficiency; thus, it may not generate samples with high rewards, particularly in the early learning stage. By definition, RL generates training samples from its model distribution. This means that, if model $p_\theta(y|x)$ has low predictive ability, only a few samples will exist with high rewards.

(iii) **Sample inefficiency.** The RL model may rarely draw samples with high rewards, which hinders to find the true gradient to optimize the objective function.

Machine translation suffers from this problem because the action (token) space is vast (typically $>10,000$ dimensions) and rewards are sparse, i.e., positive rewards are observed only at the end of a sequence. Therefore, the RL-based approach usually requires good initialization and thus is *not* self-contained. Previous studies have employed pre-training with ML before performing on-policy RL-based sampling (Ranzato et al., 2016; Bahdanau et al., 2017).

**Entropy regularized RL**  To prevent the policy from becoming overly greedy and deterministic, some studies have used the following entropy-regularized version of the policy gradient objective function (Mnih et al., 2016):

$$\mathcal{L}^*_{(\tau)}(\theta) := \sum_{x \in \mathcal{D}} \left\{ -\tau \mathbb{H}(p_\theta(y|x)) - \sum_{y \in \mathcal{Y}} p_\theta(y|x) r(y, y^*|x) \right\}. \tag{3}$$

Note that $\lim_{\tau \to 0} \mathcal{L}^*_{(\tau)} = \mathcal{L}^*$ holds.

**Reward augmented ML**  Norouzi et al. (2016) proposed RAML, which solves problems (i) and (iii) simultaneously. RAML replaces the sampling distribution of ML, i.e., $q(y|x)$ in Eq. (1), with a reward-based distribution $q_{(\tau)}(y|x) \propto \exp \{ r(y, y^*|x)/\tau \}$. In other words, RAML incorporates the reward information into the ML objective function. The RAML objective function is expressed as follows:

$$\mathcal{L}_{(\tau)}(\theta) := - \sum_{x \in \mathcal{X}} \sum_{y \in \mathcal{Y}} q_{(\tau)}(y|x) \log p_\theta(y|x). \tag{4}$$

However, problem (ii) remains.

Despite these various attempts, a fundamental technical barrier exists. This barrier prevents solving the three problems using a single method. The barrier originates from a trade-off between sampling distribution discrepancy (ii) and sample inefficiency (iii), because these issues are related to the sampling distribution. Thus, our approach is to control the trade-off of the sampling distributions by combining them.

## 3  $\alpha$-DIVERGENCE

The proposed method utilizes $\alpha$-divergence $D_{\mathrm{A}}^{(\alpha)}(p\|q)$, which measures the asymmetric distance between two distributions $p$ and $q$ (Amari, 1985). A prominent feature of $\alpha$-divergence is that it can behave as $D_{\mathrm{KL}}(p\|q)$ or $D_{\mathrm{KL}}(q\|p)$ depending on the value of $\alpha$, i.e., $D_{\mathrm{A}}^{(1)}(p\|q) := \lim_{\alpha \to 1} D_{\mathrm{A}}^{(\alpha)}(p\|q) = D_{\mathrm{KL}}(p\|q)$ and $D_{\mathrm{A}}^{(0)}(p\|q) := \lim_{\alpha \to 0} D_{\mathrm{A}}^{(\alpha)}(p\|q) = D_{\mathrm{KL}}(q\|p)$. This fact follows from the definition of $\alpha$-divergence

$$D_{\mathrm{A}}^{(\alpha)}(p\|q) := \frac{1}{\alpha(1-\alpha)} \left\{ 1 - \sum_{y \in \mathcal{Y}} p^\alpha(y) q^{1-\alpha}(y) \right\} = -\frac{1}{\alpha} \sum_{y \in \mathcal{Y}} p(y) \log_{(\alpha)} \left( \frac{q(y)}{p(y)} \right), \tag{5}$$

where $\log_{(\alpha)}(\cdot)$ is the generalized logarithm $\log_{(\alpha)}(x) := (1-\alpha)^{-1}(x^{1-\alpha} - 1)$. Furthermore, $\alpha$-divergence becomes a Hellinger distance when $\alpha$ equals to $1/2$.

## 4 PROPOSED OBJECTIVE FUNCTION: $\alpha$-DM

In this section, we describe the proposed objective function $\alpha$-DM and its gradient. Furthermore, we demonstrate that it can smoothly bridge both ML- and RL- based objective functions.

### 4.1 OBJECTIVE FUNCTION

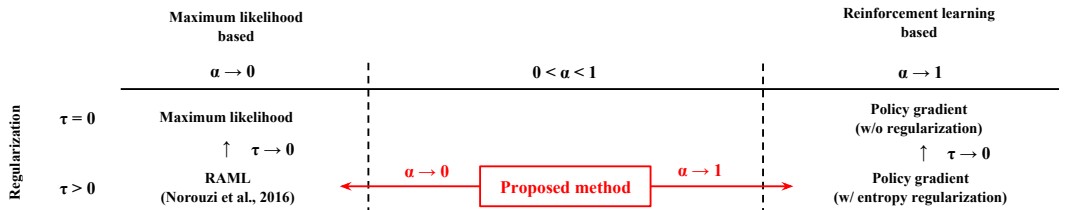

Figure 1: $\alpha$-DM objective bridges ML- and RL-based objectives.

We define the $\alpha$-DM objective function as the $\alpha$-divergence between $p_\theta$ and $q_{(\tau)}$:

$$\mathcal{L}_{(\alpha,\tau)}(\theta) := \tau \sum_{x \in \mathcal{X}} D_{\mathrm{A}}^{(\alpha)}(p_\theta \| q_{(\tau)}) = -\frac{\tau}{\alpha} \sum_{x \in \mathcal{X}} \sum_{y \in \mathcal{Y}} p_\theta(y|x) \log_{(\alpha)} \left( \frac{q_{(\tau)}(y|x)}{p_\theta(y|x)} \right). \tag{6}$$

This $\alpha$-divergence is equal to $\mathcal{L}_{(\tau)}^*$ in Eq. (3) or $\mathcal{L}_{(\tau)}$ in Eq. (4) by employing $\alpha \to 1$ or $\alpha \to 0$ limits, respectively (up to constant).

$$\lim_{\alpha \to 1} \mathcal{L}_{(\alpha,\tau)}(\theta) = \tau \sum_{x \in \mathcal{X}} D_{\mathrm{KL}}(p_\theta \| q_{(\tau)}) = \mathcal{L}_{(\tau)}^*(\theta) + \text{constant}, \tag{7}$$

$$\lim_{\alpha \to 0} \mathcal{L}_{(\alpha,\tau)}(\theta) = \tau \sum_{x \in \mathcal{X}} D_{\mathrm{KL}}(q_{(\tau)} \| p_\theta) = \tau \mathcal{L}_{(\tau)}(\theta) + \text{constant}. \tag{8}$$

Figure 1 illustrates how the $\alpha$-DM objective bridges the ML- and RL-based objective functions. Although the objectives $\mathcal{L}_{(\alpha,\tau)}^*(\theta)$, $\mathcal{L}_{(\tau)}^*(\theta)$, and $\mathcal{L}_{(\tau)}(\theta)$ have the same global minimizer $p_\theta(y|x) = q_{(\tau)}(y|x)$, empirical solutions often differ.

### 4.2 OBJECTIVE FUNCTION GRADIENT

To train neural network or other machine learning models via $\alpha$-divergence minimization, one can use the gradient of $\alpha$-DM objective function. The gradient of Eq. (6) can be expressed as

$$\nabla_\theta \mathcal{L}_{(\alpha,\tau)}(\theta) = -\sum_{x \in \mathcal{X}} \sum_{y \in \mathcal{Y}} p_\theta^{(\alpha,\tau)}(y|x) \nabla_\theta \log p_\theta(y|x), \tag{9}$$

where

$$p_\theta^{(\alpha,\tau)}(y|x) = \frac{\tau}{1-\alpha} p_\theta^\alpha(y|x) q_{(\tau)}^{1-\alpha}(y|x) \tag{10}$$

is a weight that mixes sampling distributions $p_\theta$ and $q_{(\tau)}$. This weight makes it clear that the $\alpha$-DM objective can be considered as a mixture of ML- and RL-based objective functions. See Appendix A for the derivation of this gradient. It converges to the gradient of entropy regularized RL or RAML by taking $\alpha \to 1$ or $\alpha \to 0$ limits, respectively (up to constant); i.e., $\lim_{\alpha \to 1} \nabla_\theta \mathcal{L}_{(\alpha,\tau)} = \nabla_\theta \mathcal{L}_{(\tau)}^*$ and $\lim_{\alpha \to 0} \nabla_\theta \mathcal{L}_{(\alpha,\tau)} = \tau \nabla_\theta \mathcal{L}_{(\tau)}$.

In Appendix C, we summarize all of the objective functions, gradients, and their connections.

## 5 $\alpha$-DM ANALYSIS

In this section, we characterize the difference between $\alpha$-DM objective function $\mathcal{L}_{(\alpha,\tau)}$ and the desired RL-based objective function $\mathcal{L}_{(\tau)}^*$ with respect to sup-norm. Our main claim is that, with

respect to sup-norm, the discrepancy between $\mathcal{L}_{(\alpha,\tau)}$ and $\mathcal{L}^*_{(\tau)}$ decreases linearly as $\alpha$ increases to 1. We utilize this analysis to motivate our $\alpha$-DM objective function with larger $\alpha$ if there are no concerns about the sampling inefficiency.

**Proposition 1** *Assume that $p_\theta$ has the same finite support $\mathcal{S}$ as that of $q_{(\tau)}$, and that for any $s \in \mathcal{S}$, there exists $\delta > 0$ such that $p_\theta(s) > \delta$ holds. For any $\alpha \in (0,1)$, the following holds.*

$$\sup_\theta \left| \mathcal{L}^*_{(\tau)}(\theta) - \tilde{\mathcal{L}}_{(\alpha,\tau)}(\theta) \right| \leq C_1(1-\alpha) + C_2, \tag{11}$$

*where $\tilde{\mathcal{L}}_{(\alpha,\tau)} := \alpha\mathcal{L}_{(\alpha,\tau)}$. Here, $C_1, C_2$ is universal constants irrelevant to $\alpha$.*

The following proposition immediately proves the theorem above.

**Proposition 2** *Assume that probability distribution $p$ has the same finite support $\mathcal{S}$ as that of $q$, and that for any $s \in \mathcal{S}$ there exists $\delta > 0$ such that $p(s) > \delta$ holds. For any $\alpha \in (0,1)$, the following holds.*

$$\sup_p \left| D_{\mathrm{KL}}(p\|q) - \alpha D_{\mathrm{A}}^{(\alpha)}(p\|q) \right| \leq C(1-\alpha). \tag{12}$$

*Here, $C = \max\left\{ \sup_p \left| \sum p \log^2(q/p) \right|, \sup_p \left| \sum q \log^2(q/p) \right| \right\}$.*

For the proof of the Proposition 1 and Proposition 2, see Appendix B.

## 6 Optimization of $\alpha$-DM objective function

In this paper, we employed the optimization strategy which is similar to that of RAML. We sample target sentence $y$ for each $x$ from another data augmentation distribution $q_0(y|x)$, and then estimate the gradient by importance sampling (IS). For example, we add some noise to the ground truth target sentence $y^*$ by insertion, substitution, or deletion, and the distribution $p_0(y|x)$ assigns some probability to each modified target sentence. Given samples from this proposal ditribution $p_0(y|x)$, we update the parameter using the following IS estimator

$$\nabla_\theta \mathcal{L}_{(\alpha,\tau)}(\theta) = -\sum_{x \in \mathcal{X}} \sum_{y \in \mathcal{Y}} q_0(y|x) \left( \frac{p_\theta^{(\alpha,\tau)}(y|x)}{q_0(y|x)} \right) \nabla \log p_\theta(y|x) \tag{13}$$

$$\simeq -\sum_{i=0}^{N} w_i \nabla \log p_\theta(y_i|x_i). \tag{14}$$

Here, $\{(x_1, y_1), \ldots, (x_N, y_N)\}$ are the $N$ samples from the proposal distribution $q_0(y|x)$, and $w_i$ is the importance weight which is proportional to $p_\theta^{(\alpha,\tau)}(y_i|x_i)$:

$$w_i \propto p_\theta^\alpha(y_i|x_i) q_{(\tau)}^{1-\alpha}(y_i|x_i). \tag{15}$$

Note that the difference betweene RAML and $\alpha$-DM is only this importance weight $w_i$. In RAML, $w_i$ depends only on $q_{(\tau)}(y_i|x_i)$ but not on $p_\theta(y_i|x_i)$. We normalize $w_i$ in each minibatch in order to use same hyperparameter (e.g., learning rate) as ML baseline. Thus, this estimator becomes a weighted IS estimator. A weighted IS estimator is not unbiased, yet but it has smaller variance. Also, we found that normalizing $q_{(\tau)}(y_i|x_i)$ and $p_\theta(y_i|x_i)$ in each minibatch leads to good results.

## 7 Numerical experiments

We evaluate the effectiveness of $\alpha$-DM experimentally using neural machine translation tasks. We compare the BLEU scores of ML, RAML, and the proposed $\alpha$-DM on the IWSLT'14 German–English corpus (Cettolo et al., 2014). In order to evaluate the impact of training objective function, we train the same attention-based encoder-decoder model (Bahdanau et al., 2015; Luong et al., 2015) for each objective function. Furthermore, we use the same hyperparameter (e.g., learning rate, dropout rate, and temperature $\tau$) between all the objective functions. For RAML and $\alpha$-DM, we employ a data augmentation procedure similar to that of Norouzi et al. (2016), and thus we generate samples from a data augmentation distribution $q_0(y|x)$. Note that the difference between RAML and $\alpha$-DM is only the weight $w_i$ of Eq. (14). The details of data augmentation distribution are described in Section 7.2.

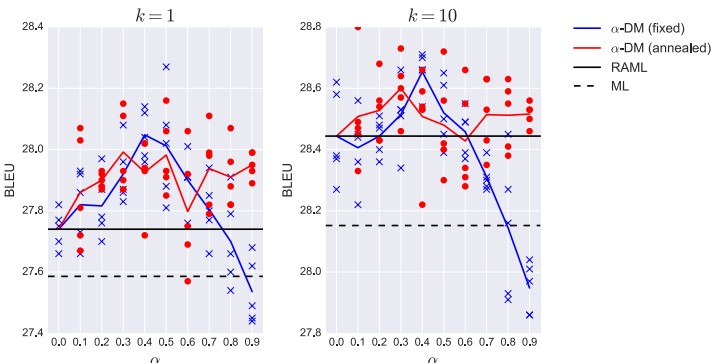

Figure 2: Performance on different hyperparameters $\alpha$. The $\alpha$-DM approach with larger $\alpha$ performs better than that with smaller $\alpha$.

Table 1: BLEU on IWSLT'14 German–English.

|  | $k = 1$ | $k = 10$ |
|---|---|---|
| ML | 27.59 ($\pm 0.18$) | 28.15 ($\pm 0.16$) |
| RAML | 27.74 ($\pm 0.06$) | 28.44 ($\pm 0.13$) |
| $\alpha$-DM, $\alpha = 0.3$ fixed | 27.92 ($\pm 0.10$) | 28.51 ($\pm 0.10$) |
| $\alpha$-DM, $\alpha = 0.4$ fixed | **28.05** ($\pm 0.07$) | **28.65** ($\pm 0.06$) |
| $\alpha$-DM, $\alpha = 0.5$ fixed | 28.01 ($\pm 0.16$) | 28.52 ($\pm 0.10$) |
| $\alpha$-DM, $\alpha = 0.3$ annealed | 27.99 ($\pm 0.11$) | 28.60 ($\pm 0.09$) |
| $\alpha$-DM, $\alpha = 0.4$ annealed | 27.93 ($\pm 0.11$) | 28.51 ($\pm 0.15$) |
| $\alpha$-DM, $\alpha = 0.5$ annealed | 27.98 ($\pm 0.11$) | 28.48 ($\pm 0.15$) |

**IWSLT'14 German–English.**  The training data comprised approximately 153K German–English sentence pairs and 7K development/test sentence pairs. The vocabulary size for the source/target were 32 009 and 22 822, respectively. The model architecture and parameters follow that of Ranzato et al. (2016) and Bahdanau et al. (2017). Specifically, we trained attention-based encoder-decoder model with the encoder of a bidirectional LSTM with 256 units and the LSTM decoder with the same number of layers and units. We exponentially decay the learning rate, and the initial learning rate is chosen using grid search to maximize the BLEU performance of ML baseline on development dataset. The important hyperparameter $\tau$ of RAML and $\alpha$-DM is also determined to maximize the BLEU performance of RAML baseline on development dataset. As a result, the initial learning rate of 0.5 and $\tau$ of 1.0 were used. Our $\alpha$-DM used the same hyperparameters as ML and RAML including the initial learning rate, $\tau$, and so on. Details about the models and parameters are discussed in Section 7.2.

## 7.1 RESULTS

To investigate the impact of hyperparameter $\alpha$, we train the neural sequence models using $\alpha$-DM 5 times for each fixed $\alpha \in \{0.0, 0.1, \ldots, 0.9\}$, and then reported the BLEU score of test dataset. Moreover, assuming that the underfitted model prevents the gradient from being stable in the early stage of training, we train the same models with $\alpha$ being linearly annealed from $0.0$ to larger values; we increase the value of $\alpha$ by adding $0.03$ at each epoch. Here, the beam width $k$ was set to 1 or 10. All BLEU scores and their averages are plotted in Figure 2. The results show that for both $k = 1, 10$, the models performance are better than smaller or larger $\alpha$ when $\alpha$ is around 0.5 ($\alpha = 0.5$). However, for larger fixed $\alpha$, the performance was worse than RAML and ML baselines. On the other hand, we can see that the annealed versions of $\alpha$-DM improve the performance of the corresponding fixed versions in relatively larger $\alpha$. As a result, in the annealed scenario, $\alpha$-DM with wide range of $\alpha \in (0, 1)$ improves on the performance consistently. This implies that the underfitted model makes the performance worse.

We summarize the average BLEU scores and their standard deviation of ML, RAML, and $\alpha$-DM with $\alpha \in \{0.3, 0.4, 0.5\}$ in Table. 1. The result shows that the BLEU score ($k = 10$) of our $\alpha$-DM outperforms ML and RAML baseline. Furthermore, although the ML baseline performances differ between our results and those of Bahdanau et al. (2017), the proposed $\alpha$-DM performance with $\alpha = 0.5$ without pre-training is comparable with the on-policy RL-based methods (Bahdanau et al., 2017). We believe that these results come from the fact that $\alpha$-DM with $\alpha > 0$ has smaller bias than that of $\alpha = 0$ (i.e., RAML).

## 7.2 DETAILS

We utilized a stochastic gradient descent with a decaying learning rate. The learning rate decays from the initial learning rate to $0.05$ with dev-decay (Wilson et al., 2017), i.e., after training each epoch, we monitored the perplexity for the development set and reduced the learning rate by multiplying it with $\delta = 0.5$ only when the perplexity for the development set does not update the best perplexity. The mini-batch size is 128. We used the dropout with probability 0.3. Gradients are rescaled when the norms exceed 5. In addition, if an unknown token, i.e., a special token representing a word that is not in the vocabulary, is generated in the predicted sentence, it was replaced by the token with the highest attention in the source sentence (Jean et al., 2015). We implemented our models using a fork from the PyTorch[1] version of the OpenNMT toolkit (Klein et al., 2017). We calculated the BLEU scores with *multi-bleu.perl*[2] script for both the development and test sets.

We obtained augmented data in the same manner as the RAML framework (Norouzi et al., 2016). For each target sentence, some tokens were replaced by other tokens in the vocabulary and we used the negative Hamming distance as reward. We assumed that Hamming distance $e$ for each sentence is less than $[m \times 0.25]$, where $m$ is the length of the sentence and $[a]$ denotes the maximum integer which is less than or equal to $a \in \mathbb{R}$. Moreover, the Hamming distance for a sample is uniformly selected from 0 to $[m \times 0.25]$. One can also use BLEU or another machine translation metric for this reward. However, we assumed proposal distribution $q_0(y|x)$ different from that of RAML. We assumed the simplified proposal distribution $q_0(y|x)$, which is a discrete uniform distribution over $[0, m \times 0.25]$. This results in hyperparameter $\tau$ used in this experiment being different from that of RAML. We search the $\tau$, which maximize the BLEU score of RAML on the development set. As a results, $\tau = 1.0$ was chosen, and $\alpha$-DM also uses this fixed $\tau$ in all the experiments.

## 8 RELATED WORKS

From the RL literature, reward-based neural sequence model training can be separated into on-policy and off-policy approaches, which differ in the sampling distributions. The proposed $\alpha$-DM approach can be considered an off-policy approach with importance sampling.

Recently, on-policy RL-based approaches for neural sequence predictions have been proposed. Ranzato et al. (2016) proposed a method that uses the REINFORCE algorithm (Williams, 1992). Based on Ranzato et al. (2016), Bahdanau et al. (2017) proposed a method that estimates a critic network and uses it to reduce the variance of the estimated gradient. Bengio et al. (2015) proposed a method that replaces some ground-truth tokens in an output sequence with generated tokens. Yu et al. (2017), Lamb et al. (2016), and Wu et al. (2017) proposed methods based on GAN (generative adversarial net) approaches (Goodfellow et al., 2014). Note that on-policy RL-based approaches can directly optimize the evaluation metric. Degris et al. (2012) proposed off-policy gradient methods using importance sampling, and the proposed $\alpha$-DM off-policy approach utilizes importance sampling to reduce the difference between the objective function and the evaluation measure when $\alpha > 0$.

As mentioned previously, the proposed $\alpha$-DM can be considered an off-policy RL-based approach in that the sampling distribution differs from the model itself. Thus, the proposed $\alpha$-DM approach has the same advantages as off-policy RL methods compared to on-policy RL methods, i.e., computational efficiency during training and learning stability. On-policy RL approaches must generate samples during training, and immediately utilize these samples. This property leads to high computational costs during training and if the model falls into a poor local minimum, it is difficult to

---

[1]http://pytorch.org
[2]https://github.com/moses-smt/mosesdecoder/blob/master/scripts/generic/multi-bleu.perl

recover from this failure. On the other hand, by exploiting data augmentation, the proposed $\alpha$-DM can collect samples before training. Moreover, because the sampling distribution is a stationary distribution independent of the model, one can expect that the learning process of $\alpha$-DM is more stable than that of on-policy RL approaches. Several other methods that compute rewards before training can be considered off-policy RL-based approaches, e.g., minimum risk training (MRT; Shen et al., 2016, RANDOMER (Guu et al., 2017), and Google neural machine translation (GNMT; Wu et al., 2016).

While the proposed approach is a mixture of ML- and RL-based approaches, this attempt is not unique. The sampling distribution of scheduled sampling (Bengio et al., 2015) is also a mixture of ML- and RL-based sampling distributions. However, the sampling distributions of scheduled sampling can differ even in the same sentence, whereas ours are sampled from a stationary distribution. To bridge the ML- and RL-based approaches, Guu et al. (2017) considered the weights of the gradients of the ML- and RL-based approaches by directly comparing both gradients. In contrast, the weights of the proposed $\alpha$-DM approach are obtained as the results of defining the $\alpha$-divergence objective function. GNMT (Wu et al., 2016) considered a mixture of ML- and RL-based objective functions by the weighted arithmetic sum of $\mathcal{L}$ and $\mathcal{L}^*$. Comparing this weighted mean objective function and $\alpha$-DM's objective function could be an interesting research direction in future.

## 9  CONCLUSION

In this study, we have proposed a new objective function as $\alpha$-divergence minimization for neural sequence model training that unifies ML- and RL-based objective functions. In addition, we proved that the gradient of the objective function is the weighted sum of the gradients of negative log-likelihoods, and that the weights are represented as a mixture of the sampling distributions of the ML- and RL-based objective functions. We demonstrated that the proposed approach outperforms the ML baseline and RAML in the IWSLT'14 machine translation task.

In this study, we focus our attention on the neural sequence generation problem, but we expect our framework may be useful to broader area of reinforcement learning. The sample inefficiency is one of major problems in reinforcement learning, and people try to mitigiate this problem by using several type of supervised learning frameworks such as imitation learning or apprenticisip learning. This alternative approaches bring another problem similar to the neural sequence generaton problem that is originated from the fact that the objective function for training is different from the one for testing. Since our framework is general and independent from the task, our approach may be useful to combine these approaches.

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

## A  GRADIENT OF $\alpha$-DM OBJECTIVE

The gradient of $\alpha$-DM can be obtained as follows:

$$\nabla_\theta \mathcal{L}_{(\alpha,\tau)}(\theta) = \nabla_\theta \left\{ -\sum_{x\in\mathcal{X}} \frac{\tau}{\alpha(1-\alpha)} \left\{ 1 - \sum_{y\in\mathcal{Y}} p_\theta^\alpha(y|x) q_{(\tau)}^{1-\alpha}(y|x) \right\} \right\} \tag{16}$$

$$= -\frac{\tau}{\alpha(1-\alpha)} \sum_{x\in\mathcal{X}} \sum_{y\in\mathcal{Y}} \nabla_\theta p_\theta^\alpha(y|x) q_{(\tau)}^{1-\alpha}(y|x) \tag{17}$$

$$= -\frac{\tau}{1-\alpha} \sum_{x\in\mathcal{X}} \sum_{y\in\mathcal{Y}} p_\theta^\alpha(y|x) q_{(\tau)}^{1-\alpha}(y|x) \nabla_\theta \log p_\theta(y|x) \tag{18}$$

$$= -\sum_{x\in\mathcal{X}} \sum_{y\in\mathcal{Y}} p_\theta^{(\alpha,\tau)}(y|x) \nabla_\theta \log p_\theta(y|x), \tag{19}$$

where

$$p_\theta^{(\alpha,\tau)}(y|x) = \frac{\tau}{1-\alpha} p_\theta^\alpha(y|x) q_{(\tau)}^{1-\alpha}(y|x). \tag{20}$$

In Eq. (18), we used the so-called *log-trick*: $\nabla_\theta p_\theta(y|x) = p_\theta(y|x) \nabla_\theta \log p_\theta(y|x)$.

# B PROOFS OF PREPOSITIONS

**Proposition 1** *Assume that probability distribution $p$ has the same finite support $\mathcal{S}$ as that of $q$, and that for any $s \in \mathcal{S}$ there exists $\delta > 0$ such that $p(s) > \delta$ holds. For any $\alpha \in (0, 1)$ the following holds.*

$$\sup_p \left| D_{\mathrm{KL}}(p\|q) - \alpha D_{\mathrm{A}}^{(\alpha)}(p\|q) \right| \leq C(1 - \alpha). \tag{21}$$

*Here, $C = \max\left\{ \sup_p \left| \sum p \log^2(q/p) \right|, \sup_p \left| \sum q \log^2(q/p) \right| \right\}$.*

*Proof.* By Taylor's theorem, there is an $\alpha' \in (\alpha, 1)$ such that

$$x^{(1-\alpha)} = 1 - \log x \cdot (\alpha - 1) + \frac{x^{(1-\alpha')}}{2} \log^2 x \cdot (\alpha - 1)^2 \tag{22}$$

Therefore,

$$D_{\mathrm{A}}^{(\alpha)}(p\|q) := -\frac{1}{\alpha} \sum p \log_{(\alpha)}\left(\frac{q}{p}\right) \tag{23}$$

$$= -\frac{1}{\alpha(1-\alpha)} \sum p \left\{ \left(\frac{q}{p}\right)^{(1-\alpha)} - 1 \right\} \tag{24}$$

$$= -\frac{1}{\alpha} \sum p \log\left(\frac{q}{p}\right) \tag{25}$$

$$\quad - \frac{(1-\alpha)}{2\alpha} \sum p \left(\frac{q}{p}\right)^{(1-\alpha')} \log^2\left(\frac{q}{p}\right) \tag{26}$$

Therefore, by Jensen's inequality we have

$$\sup_p \left| D_{\mathrm{KL}}(p\|q) - \alpha D_{\mathrm{A}}^{(\alpha)}(p\|q) \right| \tag{27}$$

$$= (1-\alpha) \sup_p \left| \sum p \left(\frac{q}{p}\right)^{(1-\alpha')} \log^2\left(\frac{q}{p}\right) \right| \tag{28}$$

$$= (1-\alpha) \sup_p \left| \sum p^{\alpha'} q^{1-\alpha'} \log^2\left(\frac{q}{p}\right) \right| \tag{29}$$

$$\leq (1-\alpha) \sup_p \left| \sum (\alpha' p + (1-\alpha')q) \log^2\left(\frac{q}{p}\right) \right| \tag{30}$$

$$\leq (1-\alpha) \left\{ \alpha' \sup_p \left| \sum p \log^2\left(\frac{q}{p}\right) \right| + (1-\alpha') \sup_p \left| \sum q \log^2\left(\frac{q}{p}\right) \right| \right\} \tag{31}$$

$$\leq (1-\alpha) \left\{ \alpha' C + (1-\alpha')C \right\} \tag{32}$$

$$= C(1-\alpha), \tag{33}$$

where $C = \max\left\{ \sup_p \left| \sum p \log^2(q/p) \right|, \sup_p \left| \sum q \log^2(q/p) \right| \right\}$.

**Proposition 2** *Assume that $p_\theta$ has the same finite support $\mathcal{S}$ as that of $q_{(\tau)}$, and that for any $s \in \mathcal{S}$, there exists $\delta > 0$ such that $p_\theta(s) > \delta$ holds. For any $\alpha \in (0, 1)$, the following holds.*

$$\sup_\theta \left| \mathcal{L}_{(\tau)}^*(\theta) - \tilde{\mathcal{L}}_{(\alpha,\tau)}(\theta) \right| \leq C_1(1-\alpha) + C_2, \tag{34}$$

*where $\tilde{\mathcal{L}}_{(\alpha,\tau)} := \alpha \mathcal{L}_{(\alpha,\tau)}$. Here, $C_1, C_2$ is universal constants irrevant to $\alpha$.*

*Proof.* Note that $\mathcal{L}^*_{(\tau)}(\theta) = \tau D_{\mathrm{KL}}(p_\theta|q_{(\tau)}) - Z(\tau)$ where $Z(\tau) :=$ $\sum_{x \in \mathcal{X}} \sum_{y \in \mathcal{Y}} \exp(r(y, y^*|x)/\tau)$. By Proposition 1 we have

$$\sup_\theta \left| \mathcal{L}^*_{(\tau)}(\theta) - \tilde{\mathcal{L}}_{(\alpha,\tau)}(\theta) \right| \tag{35}$$

$$= \left| \tau \sum_{\mathcal{X}} D_{\mathrm{KL}}(p_\theta|q_{(\tau)}) - Z(\tau) - \alpha\tau \sum_x D^\alpha_{\mathrm{A}}(p_\theta|q_{(\tau)}) \right| \tag{36}$$

$$\leq \tau \sum_{x \in \mathcal{X}} \left| D_{\mathrm{KL}}(p_\theta|q_{(\tau)}) - \alpha D^\alpha_{\mathrm{A}}(p_\theta|q_{(\tau)}) \right| + |Z(\tau)| \tag{37}$$

$$\leq C_1(1 - \alpha) + C_2, \tag{38}$$

where $C_1 = \tau \max\left\{\sup_\theta \left|\sum_{x \in \mathcal{X}} \sum_{y \in \mathcal{Y}} p_\theta \log^2(q_{(\tau)}/p_\theta)\right|, \sup_\theta \left|\sum_{x \in \mathcal{X}} \sum_{y \in \mathcal{Y}} q_{(\tau)} \log^2(q_{(\tau)}/p_\theta)\right|\right\}$ and $C_2 = |Z(\tau)|$.

## C  CATALOG OF OBJECTIVE FUNCTIONS AND THEIR GRADIENTS

In this section, we summarize the objective functions of

- ML (Maximum Likelihood),

- RL (Reinforcement Learning),

- RAML (Reward Augmented Maximum Likelihood; Norouzi et al., 2016),

- EnRL (Entropy regularized Reinforcement Learning), and

- $\alpha$-DM ($\alpha$-Divergence Minimization Training).

**Objectives.**   The objective functions of ML, RL, RAML, EnRL, and $\alpha$-DM are as follows

$$\mathcal{L}(\theta) = -\sum_{x \in \mathcal{X}} \sum_{y \in \mathcal{Y}} q(y|x) \log p_\theta(y|x), \tag{39}$$

$$\mathcal{L}^*(\theta) = -\sum_{x \in \mathcal{X}} \sum_{y \in \mathcal{Y}} p_\theta(y|x) r(y, y^*), \tag{40}$$

$$\mathcal{L}_{(\tau)}(\theta) = -\sum_{x \in \mathcal{X}} \sum_{y \in \mathcal{Y}} q_{(\tau)}(y|x) \log p_\theta(y|x), \tag{41}$$

$$\mathcal{L}^*_{(\tau)}(\theta) = -\sum_{x \in \mathcal{X}} \sum_{y \in \mathcal{Y}} p_\theta(y|x) \Big\{ r(y, y^*|x) - \tau \log p_\theta(y|x) \Big\}, \tag{42}$$

$$\mathcal{L}_{(\alpha,\tau)}(\theta) = -\frac{\tau}{\alpha(1 - \alpha)} \sum_{x \in \mathcal{X}} \sum_{y \in \mathcal{Y}} \Big\{ 1 - p^\alpha_\theta(y|x) q^{(1-\alpha)}_{(\tau)}(y|x) \Big\}, \tag{43}$$

where $q_{(\tau)}(y|x) \propto \exp\left\{r(y, y^*|x)/\tau\right\}$. Typically, $q(y|x) = \delta(y, y^*|x)$ where $y^*$ is the target with the highest reward.

We can rewrite some of these functions using KL or $\alpha$-divergences:

$$\mathcal{L}_{(\tau)}(\theta) = \sum_{x \in \mathcal{X}} D_{\mathrm{KL}}(q_{(\tau)} \| p_\theta) + \text{constant}, \tag{44}$$

$$\mathcal{L}^*_{(\tau)}(\theta) = \tau \sum_{x \in \mathcal{X}} D_{\mathrm{KL}}(p_\theta \| q_{(\tau)}) + \text{constant}, \tag{45}$$

$$\mathcal{L}_{(\alpha,\tau)}(\theta) = \tau \sum_{x \in \mathcal{X}} D^{(\alpha)}_{\mathrm{A}}(p_\theta \| q_{(\tau)}). \tag{46}$$

In the limits, there are the following connections between the objectives.

$$\lim_{\tau \to 0} \mathcal{L}_{(\tau)} = \mathcal{L}(\theta), \tag{47}$$

$$\lim_{\tau \to 0} \mathcal{L}_{(\tau)}^* = \mathcal{L}^*(\theta), \tag{48}$$

$$\lim_{\alpha \to 0} \mathcal{L}_{(\alpha,\tau)}(\theta) = \tau \mathcal{L}_{(\tau)}(\theta) + \text{constant}, \tag{49}$$

$$\lim_{\alpha \to 1} \mathcal{L}_{(\alpha,\tau)}(\theta) = \mathcal{L}_{(\tau)}^*(\theta) + \text{constant}. \tag{50}$$

**Gradients.** We list the gradient of each objective function and summarize the connections of them in the limit.

$$\nabla_\theta \mathcal{L}(\theta) = -\sum_{x \in \mathcal{X}} \sum_{y \in \mathcal{Y}} q(y|x) \nabla_\theta \log p_\theta(y|x), \tag{51}$$

$$\nabla_\theta \mathcal{L}^*(\theta) = -\sum_{x \in \mathcal{X}} \sum_{y \in \mathcal{Y}} p_\theta(y|x) r(y, y^*) \nabla_\theta \log p_\theta(y|x)), \tag{52}$$

$$\nabla_\theta \mathcal{L}_{(\tau)}(\theta) = -\sum_{x \in \mathcal{X}} \sum_{y \in \mathcal{Y}} q_{(\tau)}(y|x) \nabla_\theta \log p_\theta(y|x), \tag{53}$$

$$\nabla_\theta \mathcal{L}_{(\tau)}^*(\theta) = -\sum_{x \in \mathcal{X}} \sum_{y \in \mathcal{Y}} p_\theta(y|x) \Big\{ r(y, y^*|x) - \tau \log p_\theta(y|x) \Big\} \nabla_\theta \log p_\theta(y|x), \tag{54}$$

$$\nabla_\theta \mathcal{L}_{(\alpha,\tau)}(\theta) = -\frac{\tau}{1-\alpha} \sum_{x \in \mathcal{X}} \sum_{y \in \mathcal{Y}} p_\theta^\alpha(y|x) q_{(\tau)}^{1-\alpha}(y|x) \nabla_\theta \log p_\theta(y|x) \tag{55}$$

Each gradient corresponds to ML, RL, RAML, EnRL, and $\alpha$-DM. To derive Eq. (54), we used $\sum_{y \in \mathcal{Y}} p_\theta(y|x) \nabla_\theta \log p_\theta(y|x) = \nabla_\theta \sum_{y \in \mathcal{Y}} p_\theta(y|x) = 0$.

The following connections hold.

$$\lim_{\tau \to 0} \nabla_\theta \mathcal{L}_{(\tau)}(\theta) = \nabla_\theta \mathcal{L}(\theta) \tag{56}$$

$$\lim_{\tau \to 0} \nabla_\theta \mathcal{L}_{(\tau)}^*(\theta) = \nabla_\theta \mathcal{L}^*(\theta) \tag{57}$$

$$\lim_{\alpha \to 0} \nabla_\theta \mathcal{L}_{(\alpha,\tau)}(\theta) = \tau \nabla_\theta \mathcal{L}_{(\tau)}(\theta) \tag{58}$$

$$\lim_{\alpha \to 1} \nabla_\theta \mathcal{L}_{(\alpha,\tau)}(\theta) = \nabla_\theta \mathcal{L}_{(\tau)}^*(\theta) + \text{constant}. \tag{59}$$

Here, Eq. (59) are derived by

$$\lim_{\alpha \to 1} \nabla_\theta \mathcal{L}_{(\alpha,\tau)}(\theta) \tag{60}$$

$$= -\lim_{\alpha \to 1} \frac{\tau}{1-\alpha} \sum_{x \in \mathcal{X}} \sum_{y \in \mathcal{Y}} p_\theta^\alpha(y|x) q_{(\tau)}^{1-\alpha}(y|x) \nabla_\theta \log p_\theta(y|x) \tag{61}$$

$$= -\lim_{\alpha \to 1} \frac{\tau}{1-\alpha} \sum_{x \in \mathcal{X}} \Big\{ \sum_{y \in \mathcal{Y}} p_\theta^\alpha(y|x) q_{(\tau)}^{1-\alpha}(y|x) \nabla_\theta \log p_\theta(y|x) - \nabla_\theta \sum_{y \in \mathcal{Y}} p_\theta(y|x) \Big\} \tag{62}$$

$$= -\tau \sum_{x \in \mathcal{X}} \sum_{y \in \mathcal{Y}} p_\theta(y|x) \Bigg\{ \lim_{\alpha \to 1} \frac{1}{1-\alpha} \Big\{ \Big( \frac{q_{(\tau)}(y|x)}{p_\theta(y|x)} \Big)^{1-\alpha} - 1 \Big\} \Bigg\} \nabla_\theta \log p_\theta(y|x) \tag{63}$$

$$= -\tau \sum_{x \in \mathcal{X}} \sum_{y \in \mathcal{Y}} p_\theta(y|x) \log \Big( \frac{q_{(\tau)}(y|x)}{p_\theta(y|x)} \Big) \nabla_\theta \log p_\theta(y|x) \tag{64}$$

$$= -\sum_{x \in \mathcal{X}} \sum_{y \in \mathcal{Y}} p_\theta(y|x) \Big\{ r(y, y^*|x) - \tau \log p_\theta(y|x) \Big\} \nabla_\theta \log p_\theta(y|x) + \text{constant}. \tag{65}$$

