# OpenReview forum: "Alpha-divergence bridges maximum likelihood and reinforcement learning in neural sequence generation"
_ICLR.cc/2018/Conference — Reject_

### Official Review · AnonReviewer1 · 2017-11-19
**writing issues, missing baseline**

**Rating:** 4
**Confidence:** 5

**Review:**

The paper proposes another training objective for training neural sequence-to-sequence models. The objective is based on alpha-divergence between the true input-output distribution q and the model distribution p. The new objective generalizes  Reward-Augmented Maximum Likelihood (RAML) and entropy-regularized Reinforcement Learning (RL), to which it presumably degenerates when alpha goes to 1 or to 0 respectively.

The paper has significant writing issues. In Paragraph “Maximum Likelihood”, page 2, the formalization of the studied problem is unclear. Do X and Y denote the complete input/output spaces, or do they stand for the training set examples only?  In the former case, the statement “x is uniformly sampled from X” does not make sense because X is practically infinite. Same applies to the dirac distribution q(y|x), the true conditional distribution of outputs given inputs is multimodal even for machine translation. If X and Y were meant to refer to the training set, it would be worth mentioning the existence of the test set. Furthermore, in the same Section 2 the paper fails to mention that reinforcement learning training also does not completely correspond to the evaluation approach, at which stage greedy search or beam search is used.

The proposed method is evaluated on just one dataset. Crucially, there is no comparison to a trivial linear combination of ML and RL, which in one way or another was used in almost all prior work, including GNMT, Bahdanau et al, Ranzato et al. The paper does not argue why alpha divergence is better that the aforementioned combination method and also does not include it in the comparison.

To sum up, I can not recommend the paper to acceptance, because (a) an important baseline is missing (b) there are serious writing issues.

---

### Official Review · AnonReviewer3 · 2017-12-01
**Review of the paper: unfortunately I do not understand main points of this paper and cannot give accurate reviews**

**Rating:** 4
**Confidence:** 1

**Review:**

This paper considers a dichitomy between ML and RL based methods for sequence generation. It is argued that the ML approach has some "discrepancy" between the optimization objective and the learning objective, and the RL approach suffers from bad sample complexity. An alpha-divergence formulation is considered to combine both methods.

Unfortunately, I do not understand main points made in this paper and am thus not able to give an accurate evaluation of the technical content of this paper. I therefore have no option but to vote for reject of this paper, based on my educated guess.

Below are the points that I'm particularly confused about:

1. For the ML formulation, the paper made several particularly confusing remarks. Some of them are blatantly wrong to me. For example,

1.1 The q(.|.) distribution in Eq. (1) *cannot* really be the true distribution, because the true distribution is unknown and therefore cannot be used to construct estimators. From the context, I guess the authors mean "empirical training distribution"?

1.2 I understand that the ML objective is different from what the users really care about (e.g., blue score), but this does not seem a "discrepancy" to me. The ML estimator simply finds a parameter that is the most consistent to the observed sequences; and if it fails to perform well in some other evaluation criterion such as blue score, it simply means the model is inadequate to describe the data given, or the model class is so large that the give number of samples is insufficient, and as a result one should change his/her modeling to make it more apt to describe the data at hand. In summary, I'm not convinced that the fact that ML optimizes a different objective than the blue score is a problem with the ML estimator.

In addition, I don't see at all why this discrepancy is a discrepancy between training and testing data. As long as both of them are identically distributed, then no discrepancy exists.

1.3 In point (ii) under the maximum likelihood section, I don't understand it at all and I think both sentences are wrong. First, the model is *not* trained on the true distribution which is unknown. The model is trained on an empirical distribution whose points are sampled from the true distribution. I also don't understand why it is evaluated using p_theta; if I understand correctly, the model is evaluated on a held-out test data, which is also generated from the underlying true distribution.

2. For the RL approach, I think it is very unclear as a formulation of an estimator. For example, in Eq. (2), what is r and what is y*? It is mentioned that r is a "reward" function, but I don't know what it means and the authors should perhaps explain further. I just don't see how one obtains an estimated parameter theta from the formulation in Eq. (2), using training examples.

---

### Official Review · AnonReviewer2 · 2017-12-05
**Incremental novelty, lack of enough experimental evidence showing significance of the proposed method**

**Rating:** 4
**Confidence:** 3

**Review:**

Summary of the paper:

This paper presents a method, called \alpha-DM (the authors used this name because they are using \alpha-Divergence to measure the distance between two distributions), that addresses three important problems simultaneously:
(a) Objective score discrepancy: i.e., in ML we minimize a cost function but we measure performance using something else, e.g., minimizing cross entropy and then measuring performance using BLEU score in Machine Translation (MT).
(b) Sampling distribution discrepancy: The model is trained using samples from true distribution but evaluated using samples from the learned distribution
(c) Sample inefficiency: The RL model might rarely draw samples with high rewards which makes it difficult to compute gradients accurately for objective function’s optimization

Then the authors present the results for machine translation task and also analysis of their proposed method.

My comments / feedback:

The paper is well written and the problem addressed by the paper is an important one. My main concerns about this work are have two aspects:
(a)	Novelty
1.	The idea is a good one and is great incremental research building on the top of previous ideas. I do not agree with statements like “We demonstrate that the proposed objective function generalizes ML and RL objective functions …” that authors have made in the abstract. There is not enough evidence in the paper to validate this statement.
(b)	Experimental Results
2.	The performance of the proposed method is not significantly better than other models in MT task. I am also wondering why authors have not tried their method on at least one more task? E.g., in CNN+LSTM based image captioning, the perplexity is minimized as cost function but the performance is measured by BLEU etc.

Some minor comments:

1.	In page 2, 6th line after eq (1), “… these two problems” --> “… these three problems”
2.	In page 2, the line before the last line, “… resolbing problem” --> “… resolving problem”

---

### Decision · Program_Chairs · 2018-01-29
**ICLR 2018 Conference Acceptance Decision**

**Decision:**

Reject

**Comment:**

The reviewers agreed that this paper is not quite ready for publication at ICLR.  One of the reviewers thought the paper was well written and easy to follow while the two others said the opposite.  One of the main criticisms was issues with the composition.  The paper seems to lack a clear formal explanation of the problem and the proposed methodology.  The reviewers in general weren't convinced by the experiments, complaining about the lack of a required baseline and that the proposed method doesn't seem to significantly help in the experiment presented.

Pros:
- The proposed idea is interesting
- The problem is timely and of interest to the community
- Addresses multiple important problems at the intersection of ML and RL in sequence generation

Cons:
- Novel but somewhat incremental
- The experiments are not compelling (i.e. the results are not strong)
- A necessary baseline is missing
- Significant issues with the writing - both in terms of clarity and correctness.